

# A hybrid approach based on k-means and SVM algorithms in selection of appropriate risk assessment methods for sectors

Fatih Topaloglu

Computer Engineering/Faculty of Engineering, Malatya Turgut Ozal University, Malatya, Turkey

## ABSTRACT

Every work environment contains different types of risks and interactions between risks. Therefore, the method to be used when making a risk assessment is very important. When determining which risk assessment method (RAM) to use, there are many factors such as the types of risks in the work environment, the interactions of these risks with each other, and their distance from the employees. Although there are many RAMs available, there is no RAM that will suit all workplaces and which method to choose is the biggest question. There is no internationally accepted scale or trend on this subject. In the study, 26 sectors, 10 different RAMs and 10 criteria were determined. A hybrid approach has been designed to determine the most suitable RAMs for sectors by using k-means clustering and support vector machine (SVM) classification algorithms, which are machine learning (ML) algorithms. First, the data set was divided into subsets with the k-means algorithm. Then, the SVM algorithm was run on all subsets with different characteristics. Finally, the results of all subsets were combined to obtain the result of the entire dataset. Thus, instead of the threshold value determined for a single and large cluster affecting the entire cluster and being made mandatory for all of them, a flexible structure was created by determining separate threshold values for each sub-cluster according to their characteristics. In this way, machine support was provided by selecting the most suitable RAMs for the sectors and eliminating the administrative and software problems in the selection phase from the manpower. The first comparison result of the proposed method was found to be the hybrid method: 96.63%, k-means: 90.63 and SVM: 94.68%. In the second comparison made with five different ML algorithms, the results of the artificial neural networks (ANN): 87.44%, naive bayes (NB): 91.29%, decision trees (DT): 89.25%, random forest (RF): 81.23% and k-nearest neighbours (KNN): 85.43% were found.

# INTRODUCTION

Risk analysis and risk assessment methods (RAMs) in occupational health and safety (OHS) have been systematically applied in the world since the 1950s (*Zio, 2018*; *Aven, 2016*). The development of risk assessment methods, which are a part of risk management, has evolved

Corresponding author
Fatih Topaloglu,
fatih.topaloglu@ozal.edu.tr

according to requirements and sectors (*Ericson, 2016*). The purposes of risk analysis in OHS are to find risk centers, evaluate them, determine measures, and ensure that the measures are implemented. Analyzes are made to evaluate life, property, environment, health, and safety. Judgments and assumptions are used in risk analysis. These evaluations may contain uncertainty if they are based on incomplete information (*Ericson, 2016*). For this reason, the business needs to use the best information sources and the most appropriate RAMs (*Moraru, Babut & Cioca, 2014*; *Karimi Azari et al., 2011*).

In OHS applications, the algorithmic process to be followed for risk assessment should be defined. Risk assessments must cover all OHS-related hazards. Some legislation, standards, and guidelines may require a more detailed risk analysis for several specific potential damages (*Chemweno et al., 2015*). While a risk assessment methodology may be valid for some workplaces, it may not be valid for complex organizations (*Ford et al., 2008*; *Dey & Ogunlana, 2004*). In such complex structures, evaluation is made using alternative methods. During the risk assessment phase, decisions are made in light of the available data (*Marhavilas, Koulouriotis & Gemeni, 2011*).

The selection phase of RAMs in a business is the most important stage; making this selection incorrectly will cause material and moral losses in the business (*Markussen, 2012*; *Stromberg et al., 2017*). When creating a risk map and performing an initial hazard analysis, which methods to choose should be decided according to the business's own needs, structure, and the magnitude of its hazards (*Villa et al., 2016*). Very few research on the criteria for selecting risk assessment methods in OHS have been published to date.

Some studies were conducted for risk assessment method selection. Three risk assessment methods were evaluated for the most suitable model according to the criteria determined by experts *Karimi Azari et al. (2011)*, they presented a methodology for risk assessment technique selection in economics and capital management (*Chemweno et al., 2015*), nine risk assessment techniques used in chemical production industries were examined. and presented eight parameters to be used in weighting and comparing risks in risk assessment forms (*Khan & Abbasi, 1998*), risk assessment methods used in 62 industrial production facilities were classified and weighting criteria to be used in practice were suggested (*Tixier et al., 2002*). A comparison of eighteen risk analysis and evaluation methods was conducted (*Marhavilas, Koulouriotis & Gemeni, 2011*). In the related domains of information and communication technologies and accident prevention, a study was provided for the evaluation and selection of risk assessment techniques, and a categorization of risk assessment methods was developed (*Ford et al., 2008*), the selection of risk assessment methods and the criteria effective in this selection and weightings are presented (*Moraru, Babut & Cioca, 2014*). The best risk assessment technique for medium-sized businesses was identified after three approaches were compared and reviewed (*Guneri, Gul & Ozgurler, 2015*), 22 risk assessment techniques were classified and compared according to six criteria (*Ericson, 2016*). In addition to all of these studies, investigations (*Khan, Rathnayaka & Ahmed, 2015*; *Rausand & Haugen, 2020*; *Harms-Ringdahl, 2001*) highlight the significance of selecting the proper risk assessment technique, the features of various techniques, and their influence on risk assessment.

Decisions in assessing risks are made in the light of certain assumptions, judgments, and available data. These evaluations involve uncertainty if they are based on incomplete information. For this reason, businesses need to choose the most accurate information and RAMs. At this point, ML techniques can contribute a lot to the field of risk assessment (_Sadeghi et al., 2020_). The study provides a pioneering method for determining the most effective risk assessment methods suitable for different sectors. Combining machine learning techniques with traditional risk assessment methods, this study offers a unique approach to addressing risk management challenges across industries.

In the study, an ML-based hybrid model consisting of k-means clustering and SVM classification algorithms was proposed to enable the selection of the most suitable RAMs for sectors. Which of the 10 different RAMs is more suitable for 26 sectors was evaluated based on the 10 most effective criteria for risk, hazard, and control in OHS risk assessment applications.

The main contributions of this study can be summarized as follows:

(1) Although there are many RAMs available, there is no RAM that will suit all workplaces and which method to choose is the biggest question. This study proposes a high-performance approach for RAM selection. An ML-based approach to determine RAM for all sectors has not been proposed in the existing literature so far.

(2) In the study, the most suitable RAMs for the sectors were selected the administrative and software problems in the selection phase were eliminated from manpower, and machine support was provided. A flexible decision system is presented for RAM selection according to the characteristics of the sectors.

(3) In this study, a hybrid model was created that recommends the most appropriate risk assessment method for sectors. In the developed model, feature extraction was made with the Chi-Square method, which is accepted in the literature, in order to make the SVM algorithm faster and more accurate classification.

(4) SVM algorithm was run on the data set divided into subsets with different characteristics using the k-means algorithm. Thus, instead of the threshold value determined for a single and large cluster affecting the entire cluster and being made mandatory for all of them, a flexible structure was created by determining separate threshold values for each sub-cluster. In this way, the performance of the proposed model is increased.

(5) The proposed hybrid model achieved 99.6% performance in RAM selection classification for sectors. The hybrid model showed a higher performance than methods using different ML algorithms.

The organization of the current study is as follows: 'Materials and Method', materials and methods are presented. The results of the proposed hybrid method and experiment are presented in 'Proposed Hybrid Method'. 'Discussion and Limitations', discussion and limitations. 'Conclusion' concludes the work.

## MATERIALS AND METHOD

The proposed research methodology considered a hybrid framework that enables the identification and classification of the most appropriate risk assessment techniques for

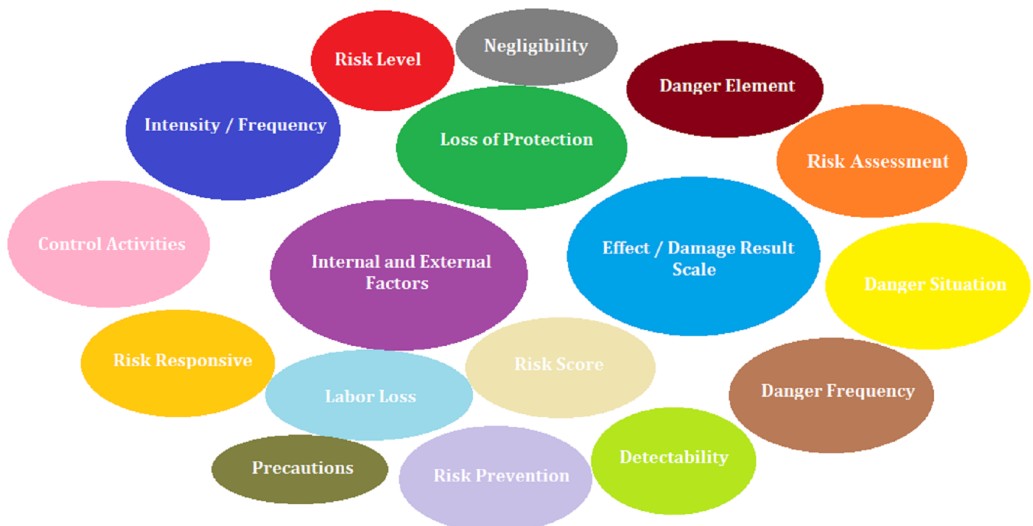

**Figure 1 Risk assessment criteria.**

sectors. A three-stage process was applied for this. First, determining the effective factors through literature review. Secondly, data preprocessing to determine the risk range with the highest performance for the sectors and to ensure the extraction of the most accurate features. Thirdly, classification and verification of the criteria determined by the hybrid method based on k-means and SVM algorithms.

## Dataset

In the study, a literature review was conducted in Scopus and Web of Science (WoS) databases to determine both the risk assessment criterion set and risk assessment methods. In this context, 223 articles were found in the WoS database and 461 articles were found in the Scopus database by using the keywords "Risk assessment methods", "occupational health and safety", "risk analysis", "risk assessment criteria". A total of 23 articles with a high level of relevance to risk assessment methods and 44 articles with a high level of relevance to risk assessment criteria were selected. As a result of the literature review, 17 technical criteria shown in Fig. 1 were determined by 9 experts with a Class A occupational safety certificate and at least 10 years of field experience.

In 'Selection of Attributes', the 10 most effective features used in practice, obtained using the chi-square feature selection method for 17 technical criteria, and the MATLAB program section of the values of these features are presented in Table 1. Of the total 260 data, 80% was used for training machine learning models and the remaining 20% was used for testing.

When we look at risk assessment methodologies, that is, methodologies and standards, all over the world within the scope of ISO 31010 Risk Management Standard for the selection of risk assessment methods, it can be seen that there are more than 150 methods. As a result of the literature review in practice, the 10 most frequently used risk assessment methods in studies were determined by the expert team.

**Table 1  Dataset cross-section.**

| Industries | Detectability | Precautions | Effect/damage result scale | Labor loss | Risk assessment | Danger frequency | Dangerous situation | Risk prevention | Risk score | Frequency |
|---|---|---|---|---|---|---|---|---|---|---|
| Justice and security | 8 | 20 | 3 | 25 | 40 | 4 | 35 | 4 | 336 | 30 |
| Mining | 3 | 80 | 8 | 85 | 70 | 9 | 95 | 7 | 810 | 90 |
| Information technologies | 7 | 30 | 3 | 35 | 30 | 5 | 40 | 3 | 389 | 40 |
| Automotive | 4 | 50 | 5 | 55 | 40 | 7 | 65 | 4 | 586 | 60 |
| Environment | 7 | 30 | 4 | 35 | 30 | 5 | 55 | 3 | 428 | 50 |

The study was based on the list of sectors determined and approved by the Vocational Qualifications Authority of the Republic of Turkey according to the sectoral qualifications included in the European Qualifications Framework consultation document adopted by the European Parliament and Council on 23 April 2008 (*Vocational Qualifications Authority, 2024*).

## K-means algorithm

MacQueen developed the k-means algorithm in 1967 (*MacQueen, 1967*). This widely used unsupervised learning method assigns each data point to only one cluster, making it a precise clustering algorithm. The method is based on the concept that the central point represents the cluster (*Han & Kamber, 2006*) and aims to find globular clusters of equal size (*Fayyad et al., 1996*).

The k-means clustering method is commonly evaluated using the sum of squared errors (SSE). The clustering result with the lowest SSE value is considered the best. To calculate SSE, sum the squares of the distances of the objects to the center points of the cluster using Eq. (1) (*Pang-Ning Tan, Steinbach & Kumar, 2006*).

$$SSE = \sum_{i=1}^{k} \sum_{x \in C_i} dist^2(m_i, x) \tag{1}$$

The objective of this criterion is to generate k clusters that are both dense and well-separated. Clustering is done by minimizing the number of nodes, n, the sum of the distances between each sensor $(x_i, y_i)$ and the cluster centroid $(X\mu i, Y\mu i)$. The distance usually used is the quadratic or Euclidean distance.

In Fig. 2, when each data point is a d-dimensional real vector represented by a set of observations (x1, x2,…,xn), K-means clustering aims to divide the n observations (k ≤ n) into clusters. The sum of squares within the cluster (minimum mean square error) within the grouped clusters (S = {S1, S2, …, Sk}) is minimized in the following Eq. (4) (*Atzori, Iera & G. Morabito, 2010*) .

$$X = \frac{1}{n}\sum_{i=1}^{n} x_i \quad Y = \frac{1}{n}\sum_{i=1}^{n} y_i \tag{2}$$

$$Cost = J = \frac{1}{n}\sum_{i=1}^{n} \left( (x_i - X_{\mu i})^2 + (y_i - Y_{\mu i})^2 \right) \tag{3}$$

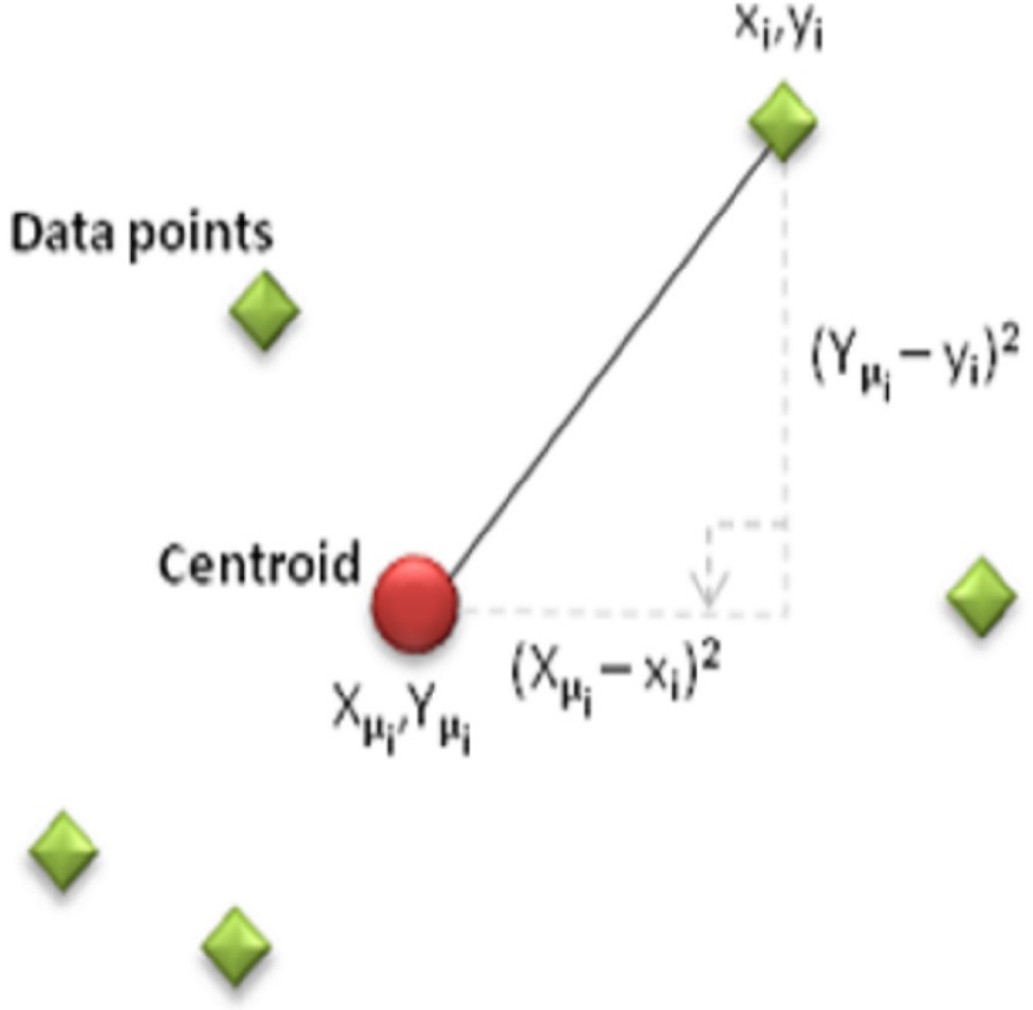

**Figure 2  Display of data points on the coordinate plane.**

$$\arg\ \min s \sum_{i=1}^{k} \sum_{x \in s_i} ||x - \mu_i||^2. \tag{4}$$

In clustering, the cost function of any centroid configuration is measured by calculating the average sum of squares of the differences between the coordinates of each data point and the nearest centroid. The aim of the algorithm is to minimize the mean square error function. The mathematical representation of clustering is shown in Fig. 3.

S: represents objects whose elements are represented by vectors, xj: represents the data set. Table 2 presents the pseudocode structure showing the mathematical interpretation of the k-means clustering algorithm.

In the study, the k-means method was used to divide the data set into subsets. The reason for this is to ensure that the SVM classification algorithm is applied not to a single

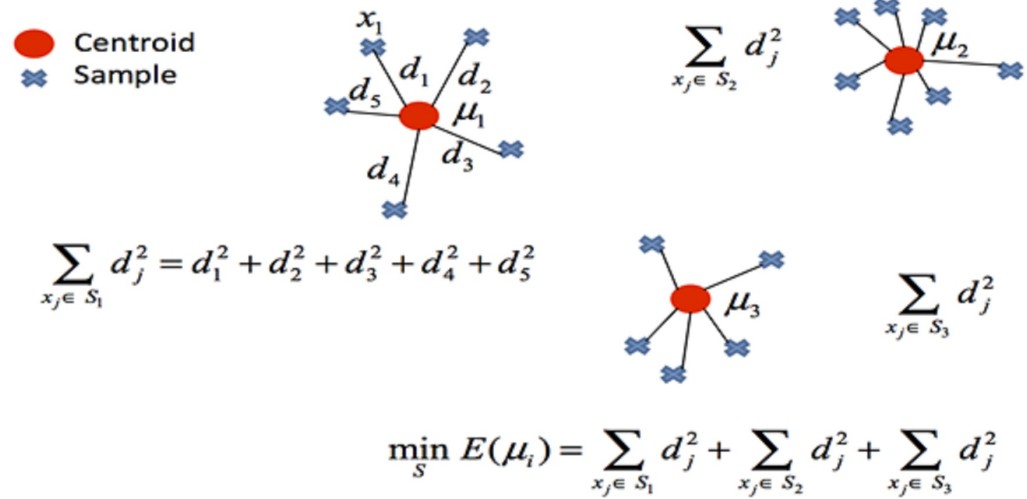

**Figure 3** Mathematical representation of clustering.

**Table 2** K-means algorithm–pseudocode.

**Input:** The weight of the document in the dataset is represented by $W_{(D)}=\{w_1, w_2, \ldots.. w_n\}$.

**Output:** The dataset is divided into clusters $C_i=\{C_1, C_2, C_3, \ldots\ldots C_n\}$.

**Begin**
**1.** Let $W_{(D)} = \{w_1, w_2, \ldots, w_n\}$ represent the weight set of the document.
$C_c = \{C_{c1}, C_{c2}, C_{c3}, \ldots, C_{cn}\}$ represent the set of cluster centres.
**2.** Select cluster centers randomly.
**3. For** all document D **do**
Clculate the distance between each data point and cluster centers using Euclidean distance
$ED \qquad = \qquad \sqrt{(W_{(D)} - C_{cn})^2}$
Assign object D to the group that is closest to centroid Cc based on a similarity measure.
**if** no documents have been moved from one group to another during the current iteration.
**then**
Stop end exit.
**else**
Recalculate the cluster centre.
$C_{cen} \qquad = \qquad \frac{1}{M}\sum_{D=1}^{M} W_{(D)}$
**end if**
**4. End for**
**End**

and large cluster, but to all subsets with different characteristics. Thus, instead of requiring a single threshold value for the entire cluster, a flexible structure is provided by applying it to subsets with different characteristics.

## Support vector machine algorithm

Support vector machines (SVM) is a supervised machine learning algorithm with strong foundations based on Vapnik–Chervonenkis theory. SVM; Although it is similar to neural networks and radial basis artificial neural networks, it generally outperforms these algorithms. SVMs are widely used in real-life classification applications. Compared to other

methods, significant improvements have been made in ease of calculation, scalability and resistance to outliers. SVM performs well on classification and regression problems even when there is a small amount of training data and a large number of features.

SVM is divided into two as Linear SVM and Non-Linear SVM according to the linear separation of data.

### Linear support vector machines

Where X and Y are subsets of $R^d$ and d is the number of features, if there is a hyperplane that can separate the elements of X and Y into different sides, X and Y can be linearly separated from each other (*Elizondo, 2006*). There are two types of SVMs, Hard-Marijn and Soft-Marjin, which are used for linearly separable situations.

*Hard-margin support vector machines.* Let the target variable be entered in two options such that the training data is $\{x_i, y_i\}$, $i = 1, 2, \ldots, L$, $y_i \in \{-1, 1\}$, $x_i \in R^d$. The formula of the hyperplane that can linearly separate the positive and negative training data from each other is as in Eq. (5). In this equation, w is the normal of the hyperplane, $|b| / ||w||$ is the perpendicular distance from the hyperplane to the origin, $||w||$ is the Euclidean norm of w (*Burges, 1998*).

$$w.x + b = 0. \tag{5}$$

The purpose of SVMs is; The aim is to keep the hyperplane separating the data belonging to different classes as far away from the closest points of all classes to each other. SVM can actually be summarized as the selection of $w$ and $b$. In this case, the data set can be defined as in Eq. (6) and summarized as in Eq. (7) (*Kartal & Balaban, 2019*).

$$x_i.w + b \begin{cases} \geq +1, & y_i = +1 \\ \leq -1, & y_i = -1 \end{cases} \tag{6}$$

$$y_i(x_i.w + b) - 1 \geq 0 \; for \; \forall i. \tag{7}$$

The planes marked $H_1$ and $H_2$ in Fig. 4 are support planes. The closest different class members on these planes are support vectors. The dividing plane passes right through the middle of the support planes. $d_1$ and $d_2$ are the distances of the support planes to the separating plane, and these two distances are equal to each other. Additionally, the sum of these distances is the margin. This margin needs to be maximized so that the dividing plane is as far away from the support vectors as possible. The margin is represented as $1 / ||w||$ based on vector geometry, and $||w||$ must be minimized to maximize the margin.

*Soft-margin support vector machines.* Soft-margin SVM is used when the training data cannot be classified without error. In order to perform the classification process with the least error, incorrectly classified data are removed from the training data set (*Cortes & Vapnik, 1995*). The Soft-Margin SVM structure showing the situations that can be classified as linear with a certain error is as shown in Fig. 5.

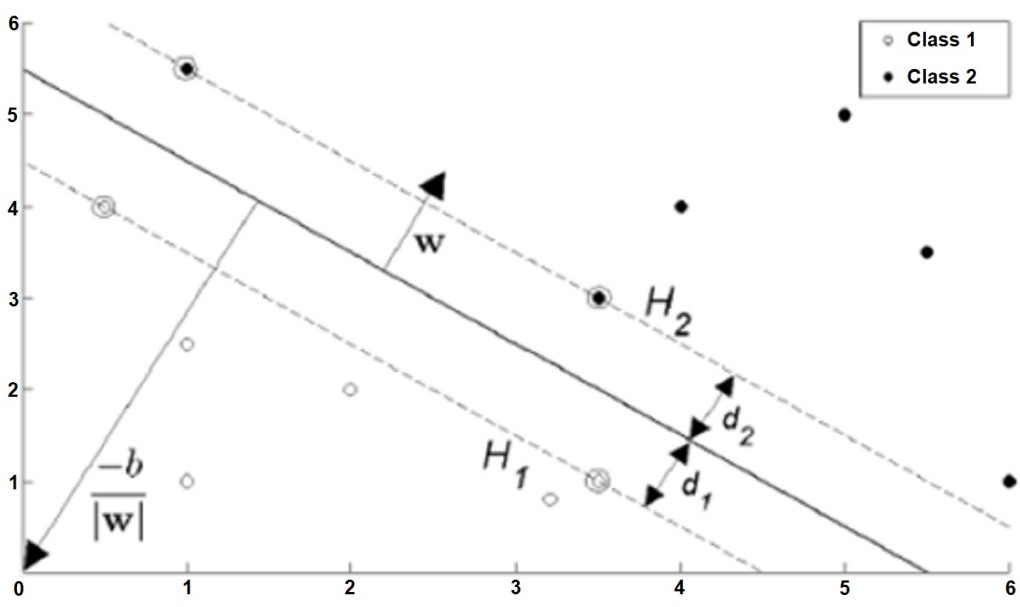

**Figure 4** Situations that can be classified linearly (hard-margin SVM) (*Fletcher, 2009*).

In soft-margin SVMs, a non-negative idle variable is defined to express Eq. (7). In this case, where $\xi i \geq 0$, Eq. (7) can be generalized as follows:

$$y_i(x_i.w+b)-1+\xi_i \geq 0 \; for \; \forall i. \tag{8}$$

The choice of $\xi i$ requires considering many situations together, and the choices affect the margin. A parameter $C$ is used to ensure the balance between $\xi i$ and margin. If $C$ is chosen larger, there will be less misclassified data. But it also causes the $wTw$ product to be large and the margin to be small. The primal problem, defined as in Eq. (7) for hard-margin, is expressed as in Eq. (9) for soft-margin. Afterwards, the processes of obtaining the dual form Eq. (10) and finding $b$ and $w$ proceed similarly to hard-margin support vector machines (*Kartal & Balaban, 2019*).

$y_i(x_i.w+b)-1+\xi_i \geq 0$ for $\forall i$ including:

$$min\frac{1}{2}||w||^2+C\sum_{i=1}^{L}\xi_i \tag{9}$$

$\sum_{i=1}^{L}a_iy_i=0, 0 \leq a_i \leq C$ for $\forall i$ including:

$$L_D \equiv \sum_{I=1}^{L}a_i - \frac{1}{2}\left(\sum_{i,j=1}^{L}a_ia_jy_iy_jx_i.x_j\right). \tag{10}$$

### Nonlinear support vector machine

Most real-life problems do not lend themselves to being separated by a linear hyperplane. In this case, SVMs map the input space to a higher dimensional space. Thus, a linear decision

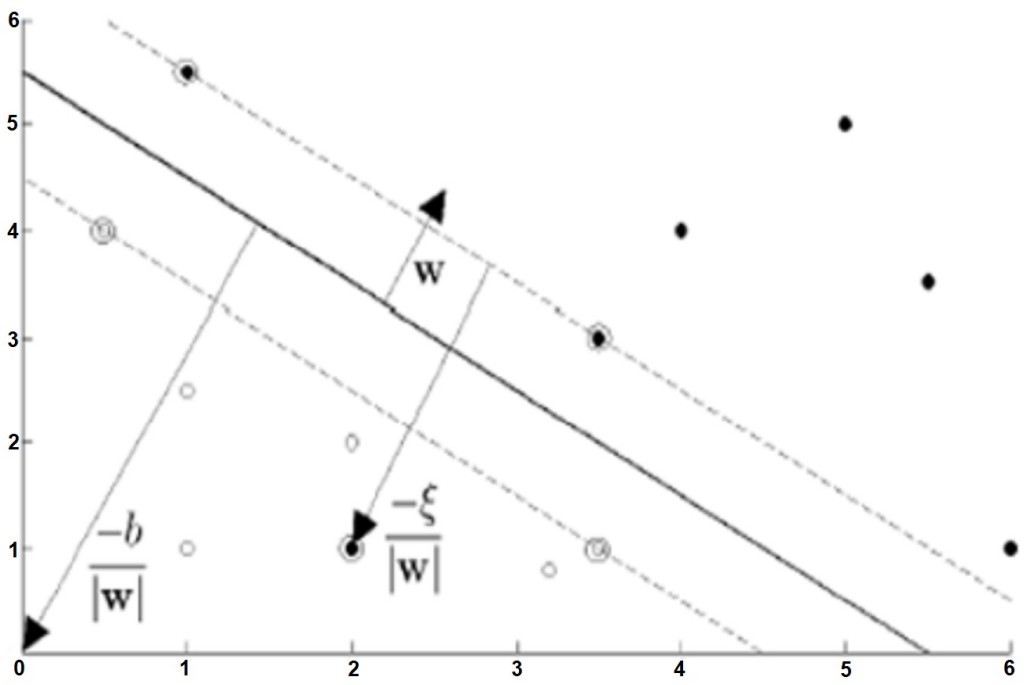

**Figure 5** Situations that can be classified as linear with a certain error (soft-margin DVM) (*Fletcher, 2009*).

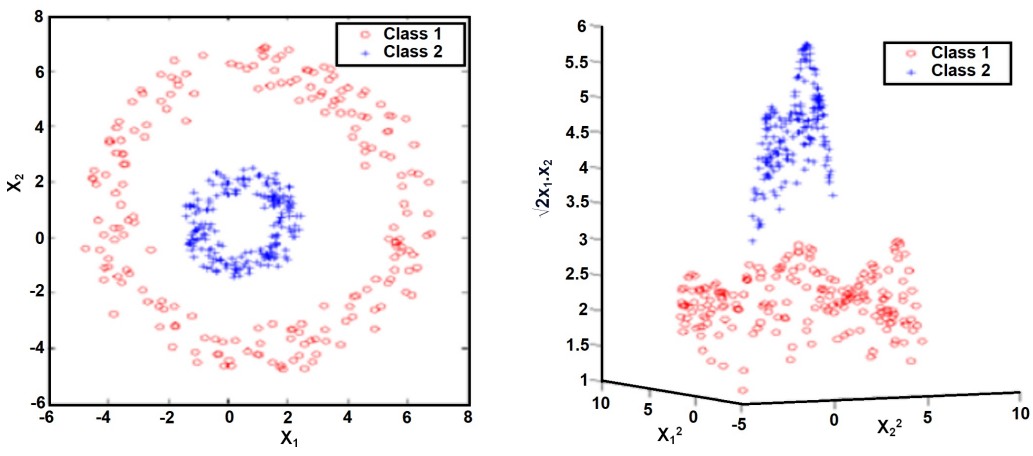

**Figure 6** Effect of kernel functions in cases that cannot be classified as linear (*Fletcher, 2009*).

boundary can be created between classes, as seen in Fig. 6. For this, the observation vector $x \in R^n$ is transformed into the vector $z \in R^F$ in a higher order space. To map $R^n \rightarrow R^F$, the $\varnothing$ function is expressed as $z = \varnothing(x)$.

$$x \in R^n \rightarrow z(x) = [a_1, \varnothing_1(x), \ldots, a_n, \varnothing_n(x)]^T \epsilon R^F. \tag{11}$$
Since the mapping function that enables this size change in linearly inseparable cases is not known and it is difficult to perform operations in high dimensions, arrangements called kernel tricks are made. Thus, instead of the mapping function in the transformed space, kernel functions that directly use the data in the input space are included in the process. Although there are many kernel functions in the literature; The most frequently used ones are linear function, polynomial function, sigmoid function and radial basis functions. Vapnik stated that the performance of these kernel functions did not make a big difference experimentally. The important thing is to determine the parameters of the selected core function (*Erasto, 2001*).

Formulations of frequently used kernel functions are as follows:

$$\text{Linear function} : K\left(x_i, x_j\right) = x_i^T x_j \tag{12}$$

$$\text{Polynomial function} : K\left(x_i, x_j\right) = \left(x_i, x_j\right)^d \tag{13}$$

$$\text{Sigmoid function} : K\left(x_i, x_j\right) = \tanh(kx_i, x_j - \delta) \tag{14}$$

$$\text{Radial basis function} : K\left(x_i, x_j\right) = \exp\left(-\gamma \left|\left|x_i - x_j\right|\right|^2\right), \gamma > 0 \tag{15}$$

The $K(x_i.x_j)$ function is called the kernel function and gives the one-to-one product of the feature space maps of the actual data points. For this reason, all elements of the dataset should be used for training. Thus, a more accurate error rate is obtained compared to manual selections. But due to random selection, there may be very small differences in error rates. Table 3 gives the pseudo-code structure of the SVM classification algorithm.

## Risk assessment methods

Risk assessment is the process of estimating the magnitude of risks arising from hazards in any system and deciding whether these risks are acceptable, taking into account the adequacy of existing controls, and can be symbolized in Fig. 7.

When we look at risk assessment methodologies, that is, methodologies and standards, all over the world, we see that there are more than 150 methods (*Özkılıç, 2007*). The most important difference between risk assessment methods is the unique methods they use to find the risk value. The advantages and disadvantages of these methods compared to each other are given in detail in the sources. The basic risk assessment methods to be analyzed for the sectors within the scope of the study and the symbols used in the study are given in Table 4.

Risk assessment techniques can be divided into two main groups in terms of estimating risks, their probability of occurrence, and their possible effects. These are qualitative and quantitative approaches (*Özkılıç, 2005*). Depending on the conditions to be analyzed, semi-quantitative methods, in which both methods are used together, can also be used.

**Table 3  SVM algorithm –pseudocode.**

**Data:** Data set containing p* variables and binary results

**Output:** Ranking of variables by relevance

Find the optimal values for SVM model tuning parameters;
Train the SVM model;
p ←p*;
**while** p ≥2 **do**
SVMp ← SVM with optimised tuning parameters for the p variables
and observations in **Data**;
$\omega p$      ←      *Calculate SVMp weight vector* $(\omega\_p1,\ldots.\omega\_pp)$;
rank.criteria ← $\left(\omega_{p1}^2,\ldots.\omega_{pp}^2\right)$;
minimum rank criterion ← variable with the lowest value in the rank criterion vector;
Remove minimum ranking criteria from **data**;
$Rank_p$ ← min.rank.criteria;
P ← p-1;
**end**
$Rank1$ ← variable in Data $\notin (Rank_2,\ldots.Rank_{p*})$;
**return** $(Rank_1,\ldots.Rank_{p*})$

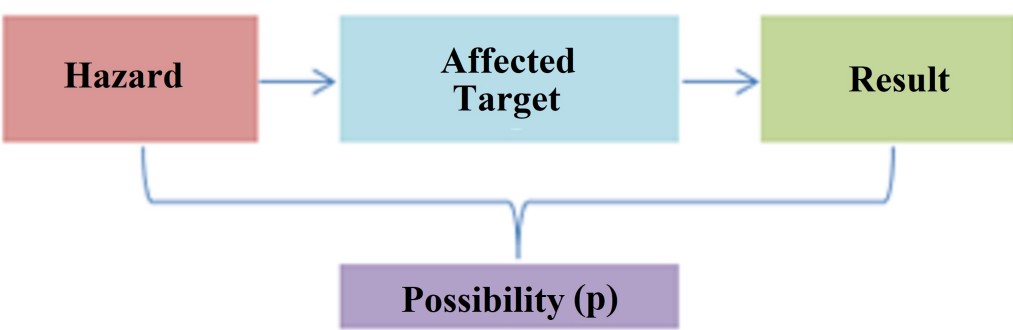

**Figure 7  Risk assessment concept.**

In quantitative risk analysis, numerical methods are used to calculate the risk. Numerical values are used as data regarding the probability of the risk occurring and what its impact will be after it occurs (*Özkılıç, 2005*). These values are analyzed with mathematical and logical methods and risk scores are found. In qualitative risk analyses, instead of numerical values, descriptive expressions such as low, and high or definitions such as A, B, I, and II are used in calculating the risk (*Özkılıç, 2005*).

These methods are part of risk management, almost the core part. In the study, 10 different risk assessment methods determined by the expert team as a result of literature review were tested for all sectors.

## PROPOSED HYBRID METHOD

In the study, a hybrid model was proposed in which k-means clustering and SVM classification algorithms, which are machine learning methods, are used together to select the most appropriate risk assessment methods that can be used in occupational health and

**Table 4  List of sectors (*Vocational Qualifications Authority, 2024*).**

| | Sector list and codes | | |
|---|---|---|---|
| **S1** | Justice and Security | **S14** | Culture, Art and Design |
| **S2** | Mining | **S15** | Woodworking, Paper and Paper Products |
| **S3** | Information Technologies | **S16** | Media Communication and Broadcasting |
| **S4** | Automotive | **S17** | Chemical, Petroleum, Rubber and Plastic |
| **S5** | Environment | **S18** | Glass, Cement and Soil |
| **S6** | Education | **S19** | Health and Social Facilities |
| **S7** | Electrical and Electronics | **S20** | Sports and Recreation |
| **S8** | Energy | **S21** | Agriculture, Hunting and Fishing |
| **S9** | Finance | **S22** | Textile, Ready-to-Wear, Leather |
| **S10** | Food | **S23** | Commerce(Sales and Marketing) |
| **S11** | Build | **S24** | Social and Personal Services |
| **S12** | Business and Management | **S25** | Tourism and Accommodation Services |
| **S13** | Metal | **S26** | Transport, Logistics and Communication |

Notes.
* The European Qualifications Framework (EQF) consultation document refers to sectoral qualifications.
* The term sector is used to describe categories on which companies base their economic activities, products or technologies (chemistry, tourism, *etc.*) or cross/horizontal professional categories (IT, marketing, banking, *etc.*).

safety applications for sectors and businesses. The block diagram of the proposed hybrid method is presented in Fig. 8.

## Data preprocessing

Risk value segmentation in the preprocessing stage was based on the Fine-Kinney method risk scale. As seen in Fig. 9, as a preliminary process, risk dimensions were evaluated by dividing them into different value ranges (<20), (20<R<70), …., (>400). The purpose of using different segments is to determine the risk range that has the highest performance for the sectors and to ensure that the most accurate features are extracted.

The study was based on the list of sectors determined and approved by the Vocational Qualifications Authority of the Republic of Turkey, according to the sectoral qualifications included in the European Qualifications Framework consultation document adopted by the European Parliament and Council on 23 April 2008. The list of sectors determined by the Vocational Qualifications Authority and to be used within the scope of the study is presented in Table 4 (*Vocational Qualifications Authority, 2024*).

When we look at risk assessment methodologies, that is, methodologies and standards, all over the world within the scope of ISO 31010 Risk Management Standard for the selection of risk assessment methods, it can be seen that there are more than 150 methods. The 10 risk assessment methods most frequently used in the studies, determined by the expert team as a result of the literature review in the study, are presented in Table 5.

## Selection of attributes

In practice, 17 criteria determined by the expert team as a result of the literature review were evaluated with the chi-square feature selection method and the most effective features were determined. The use of unimportant attributes that will not affect the result in the

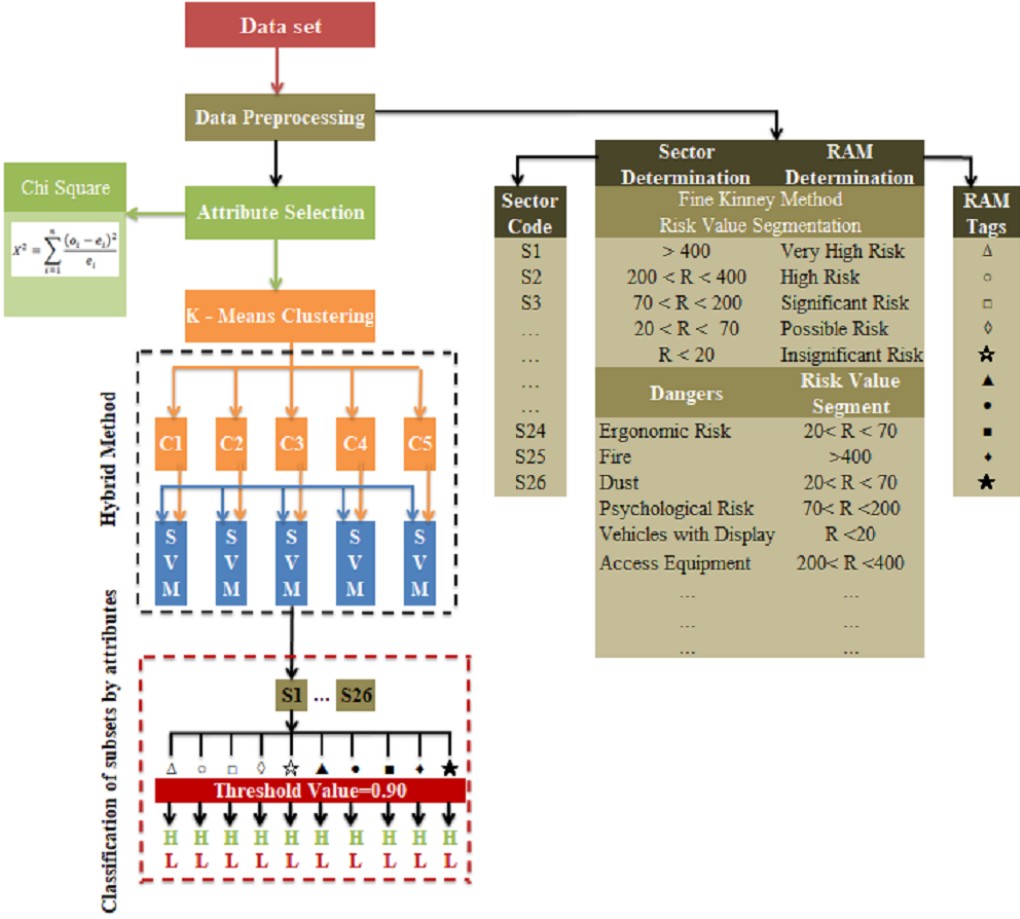

**Figure 8** Block diagram of the proposed hybrid method.

selection of the most appropriate risk assessment method increases the processing time and may also cause a decrease in performance (*Chebrolu, Abraham & Thomas, 2005*). The chi-square method, also known as the $X^2$ test, can be used to determine whether the variables are suitable for describing the dataset (*Kavzoğlu, Şahin & Çölkesen, 2014*).

There are two hypotheses in the chi-square test: $H_0$ and $H_1$. $H_0$ is the hypothesis that the variables in the data set are appropriate, and $H_1$ is the hypothesis that the variables in the data set are not suitable. If the calculated value is greater than the determined value, the $H_1$ hypothesis is accepted, and if it is smaller, the $H_0$ hypothesis is accepted. How the chi-square statistic is calculated is formulated in Eq. (16).

$$X^2 = \sum_{i=1}^{n} \frac{(o_i - e_i)^2}{e_i} \tag{16}$$

In this equation, n represents the number of features in the data set, $o_i$ represents the observed frequency value for the ith feature, and $e_i$ represents the expected frequency value

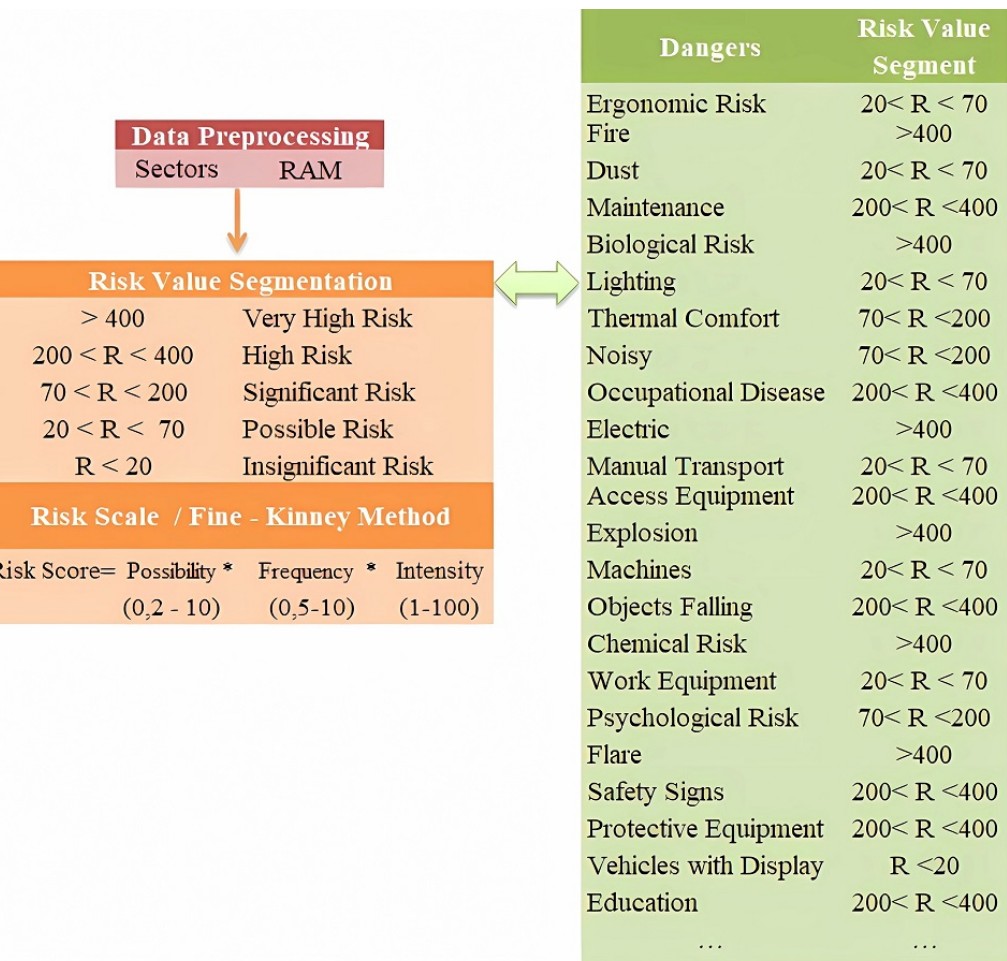

**Figure 9** Fine-Kinney method risk scale.

**Table 5 Risk assessment methods and symbols.**

| Classification tags | Risk assessment methods |
| --- | --- |
| △ | L Type Matrix Analysis |
| ◯ | X Type Matrix Analysis |
| □ | Fine-Kinney Method |
| ◇ | Failure Modes and Effects Analysis (FMEA) |
| ☆ | Preliminary Hazard Analysis (PHA) |
| ▲ | Fault Tree Analysis (FTA) |
| • | Hazard and Operability Analysis (HAZOP) |
| ■ | Event Tree Analysis (ETA) |
| ♦ | What If |
| ★ | Job Security Analysis (JSA) |

for the ith feature. The features obtained with the chi square algorithm within the scope of the study are given in Fig. 10.

The features obtained using the chi-square feature selection method for 17 technical criteria defined in the data set were found as Precautions, Risk Assessment, Risk Prevention, Intensity/Frequency, Detectability, Effect/Damage Result Scale, Labor Loss, Danger Frequency, Danger Situation and Risk Score.

## Hybrid method

In the k-means method, which is the first step of the application, the analysis results made on the data set with k = {3, 4, 5, 6, 7} values were examined to select the most appropriate {k}. K-means code structure in Matlab application in Table 6:

>>[idx,ort,sumd,D] =kmeans(Dataset_Name, **Cluster_Number**);

When the results obtained with were examined, it was seen that the best result was obtained for $k = 5$ with an accuracy of 0.9763 and an error rate of 0.0153. Additionally, the Matlab program analysis images obtained for $k = 5$ are presented in Fig. 11.

In the second step of the application, the SVM classification algorithm is run for each of the five subsets formed by k-means application, and the risk assessment methods whose average is greater than the determined threshold value are classified as appropriate, while those less than the threshold value are classified as unsuitable. All subsets were run for threshold value = {0.80,0.85,0.90, 0.95} and the threshold value that gave the best results was selected. SVM code structure in Matlab application:

>>pt = cvpartition(Dataset_Name,"HoldOut",0.2);
hdTrain = Data(training(pt),:);
hdTest = Data(test(pt),:);
>>svmModel = fitcsvm(hdTrain",Dataset_Ad ı");
>>tahmin = predict(svmModel,hdTest)
>>hata = loss(svmModel,hdTest);

The training data results of the SVM application for $k = 5$ obtained with are shown in Table 7, the test data results are shown in Table 8, and the ROC curves representing all threshold values are shown in Fig. 12.

When the ROC curves and accuracy metrics in the tables were examined, it was seen that the best classification result was obtained for the threshold value = 0.90 in the data set.

In the third step of the application, the experimental setup and results obtained by running the optimum number of clusters determined by the k-means clustering algorithm for $k = 5$ and the optimum threshold value obtained by running the SVM classification algorithm on all clusters for TV = 0.90 are presented in Fig. 13.

In the study, the detailed results of the experimental setup designed in Fig. 14 are as follows:

(a), the data set codes determined by the Vocational Qualifications Authority and representing the 26 sectors used within the scope of the study in a single and large cluster are shown.

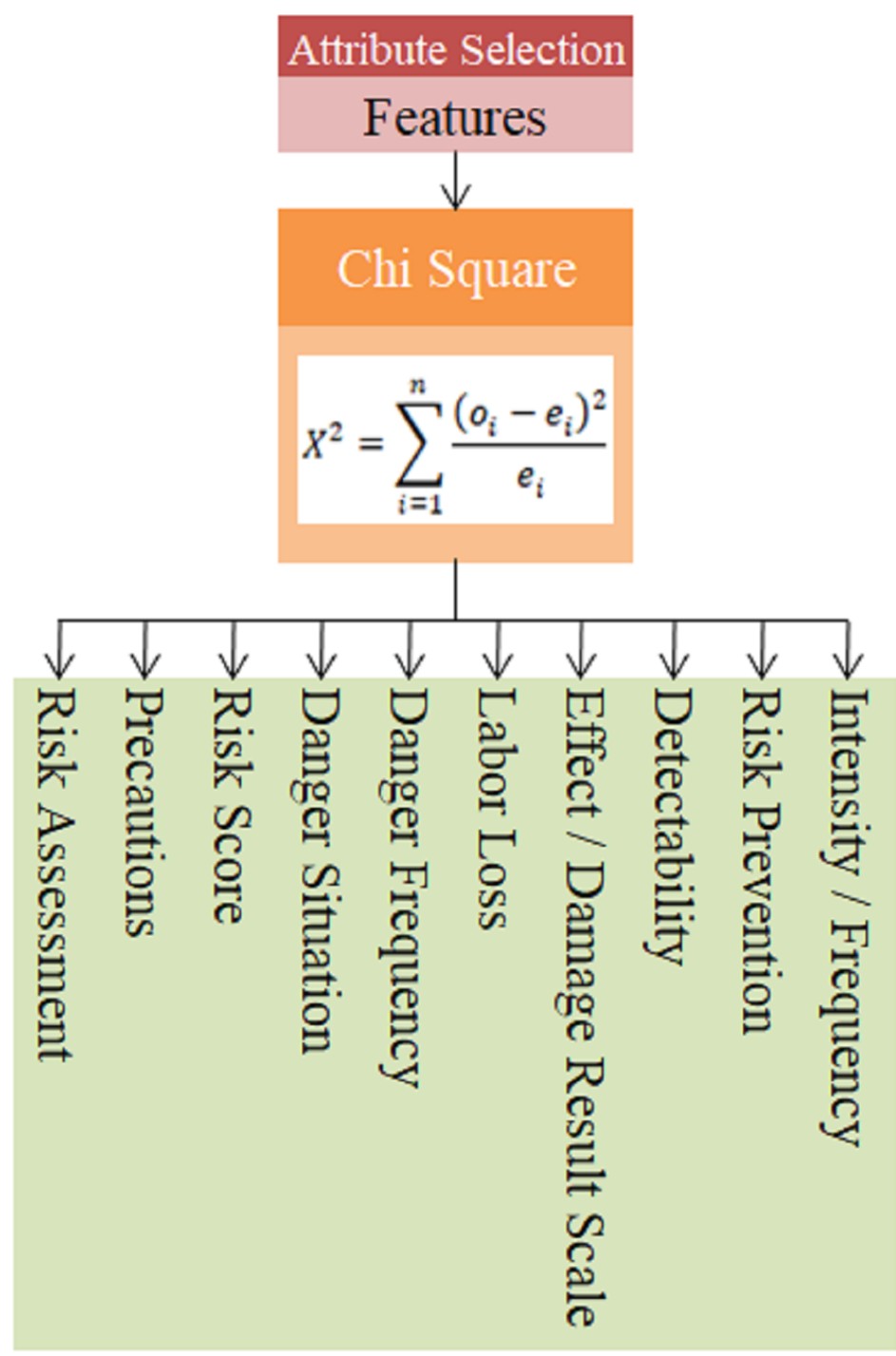

**Figure 10 Chi square method feature output.**

**Table 6  K-means analysis results.**

| k | 3 | 4 | 5 | 6 | 7 |
|---|---|---|---|---|---|
| **Accuracy** | 0.7236 | 0.7589 | 0.9763 | 0.9451 | 0.9417 |
| **Error** | 0.2143 | 0.1927 | 0.0153 | 0.0246 | 0.0267 |
| **Precision** | 0.8901 | 0.9341 | 0.9865 | 0.9776 | 0.9705 |
| **Recall** | 0.8908 | 0.9448 | 0.9775 | 0.9771 | 0.9695 |

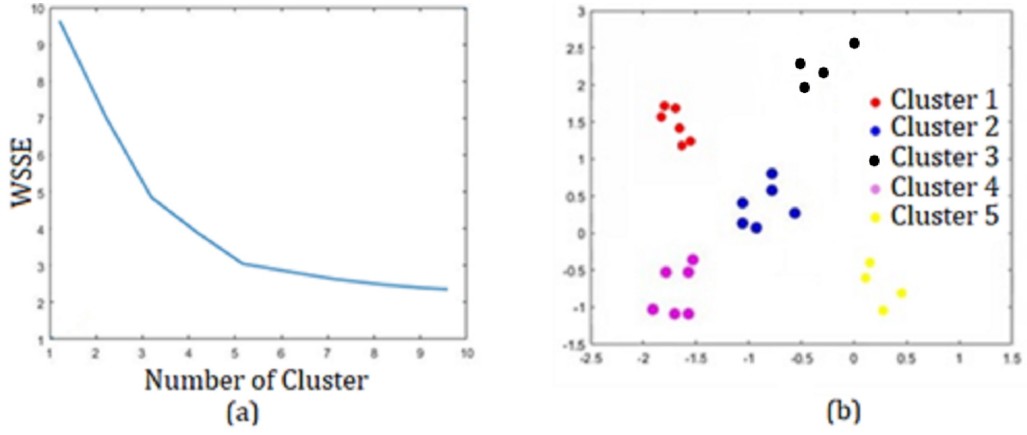

**Figure 11  For $k = 5$ (A) Elbow method. (B) Sector segmentation results.**

**Table 7  SVM algorithm training data results.**

| Threshold values (%) | Training data set | | | |
|---|---|---|---|---|
| | Accuracy | Recall | F1 Score | Precision |
| 0.95 | 0.8970 | 0.8837 | 0.8614 | 0.8837 |
| 0.90 | 0.9946 | 0.9704 | 0.9448 | 0.9716 |
| 0.85 | 0.9688 | 0.9565 | 0.9341 | 0.9567 |
| 0.80 | 0.9597 | 0.9413 | 0.9128 | 0.9424 |

(b), in the k-means clustering method, sectors are grouped for the optimum number of clusters $k = 5$, determined using the Euclidean distance algorithm. Thus, instead of the SVM classification algorithm affecting the entire cluster and being required for all of them, the infrastructure for applying it separately for each subset separated according to their characteristics is provided. For example, cluster C1; from Automotive (S4), Electrical and Electronics (S7), Wood Processing/Paper/Paper Products (S15), Textile/Ready-made Clothing/Paper Products (S22), Tourism and Accommodation (S25) and Transportation/Logistics/Communication (S26) sectors is formed.

(c), the most appropriate risk assessment methods were found for each subset and the sectors in this cluster with SVM. For example, for cluster C1; Fine–Kinney Method, X Type Matrix Analysis, Preliminary Hazard Analysis (PHA), Fault Tree Analysis (FTA), What if

**Table 8  SVM algorithm test data results.**

| Threshold values (%) | Test data set | | | |
|---|---|---|---|---|
| | Accuracy | Recall | F1 Score | Precision |
| 0.95 | 0.7801 | 0.7512 | 0.7341 | 0.7512 |
| 0.90 | 0.9445 | 0.9345 | 0.9153 | 0.9345 |
| 0.85 | 0.9262 | 0.9010 | 0.8818 | 0.9010 |
| 0.80 | 0.8921 | 0.8739 | 0.8661 | 0.8739 |

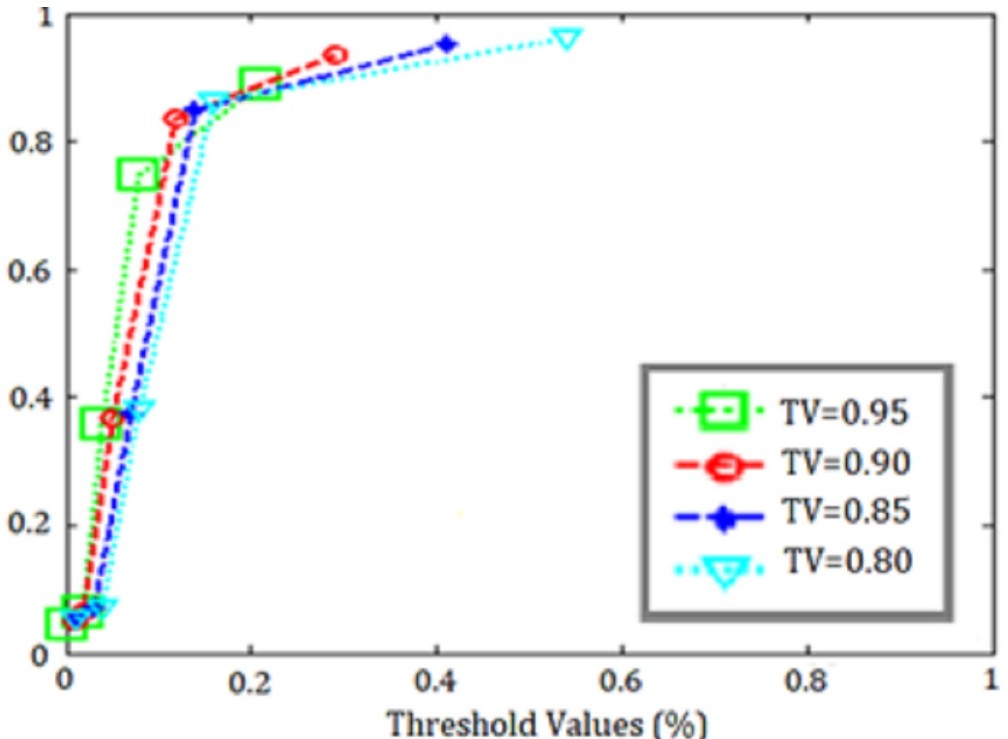

**Figure 12  ROC curves for SVM threshold values.**

and Job Security Analysis (JSA) were found to be the most appropriate risk assessment methods.

(d), SVM was run for the threshold value {0.80,0.85,0.90, 0.95} for the C1 subset and the best threshold value was selected. The optimum threshold value for the C1 subset was found to be = 0.95. This process was repeated for each subset and the optimum threshold value for all clusters was found to be TV = 0.90, as seen in Table 7.

(e), k-means and SVM hybrid methods were run for the optimum threshold value determined for each cluster, and the most appropriate final risk assessment methods were

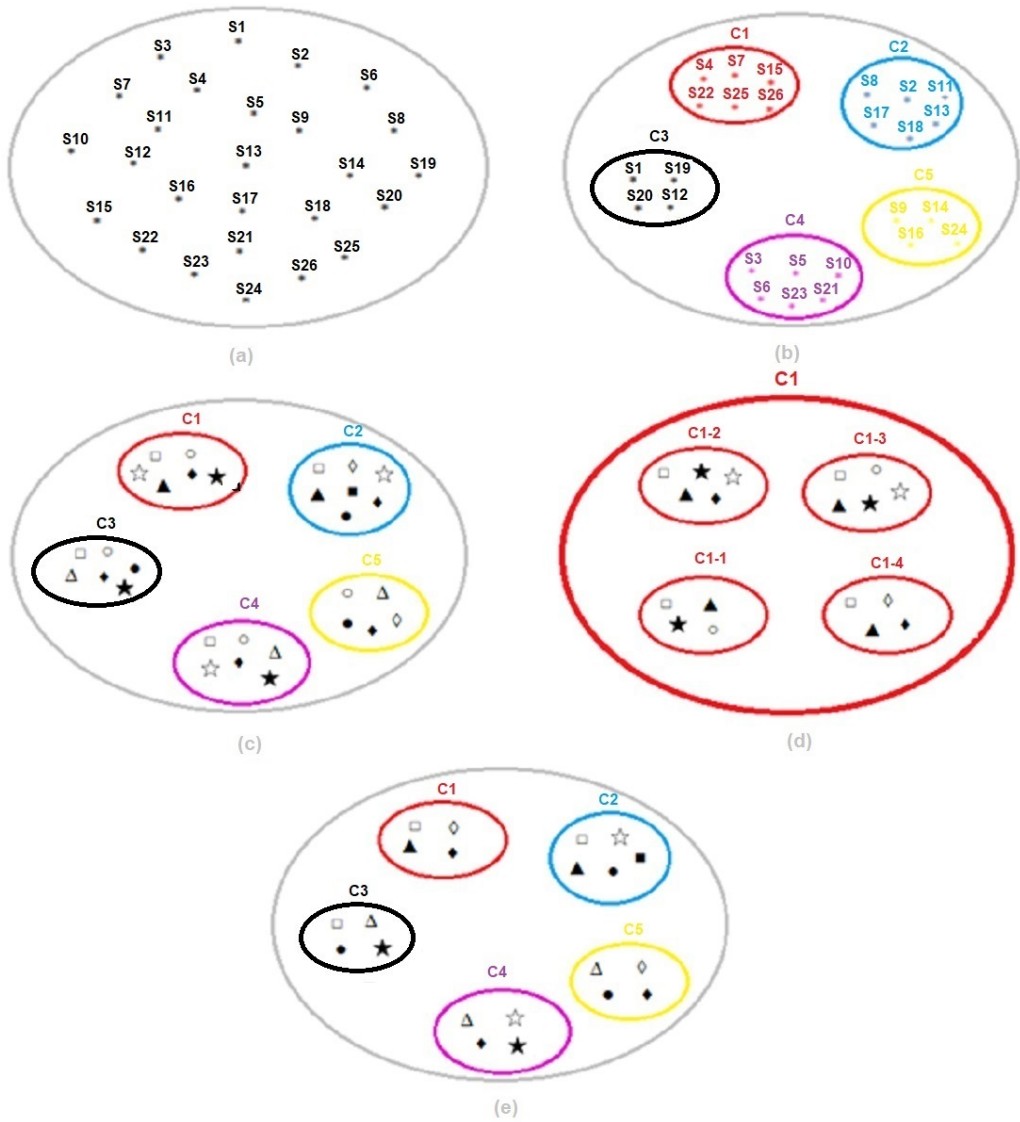

**Figure 13** (A) Dataset. (B) Segmentation of the dataset with k-means. (C) Classification of each subset with SVM. (D) Selection of the best threshold value for each subset with SVM. (E) Result of k-means and SVM hybrid method.

determined. In this context, the final result for cluster C1; Fine Kiney Method, Fault Tree Analysis (FTA), What if and Failure Models and Effects Analysis (FMEA) were found.

Table 9 presents the performance results of the k-means algorithm, SVM algorithm and the proposed hybrid method (k-means and SVM) algorithm.

Comparative performance results of the proposed hybrid method with artificial neural networks (ANN), naive Bayes (NB), decision tree (DT), random forest (RF) and k-nearest neighbors (KNN) machine learning algorithm are presented in Table 10. Confusion matrix results are presented in Fig. 14. Additionally, $R^2$, mean absolute error (MAE), mean

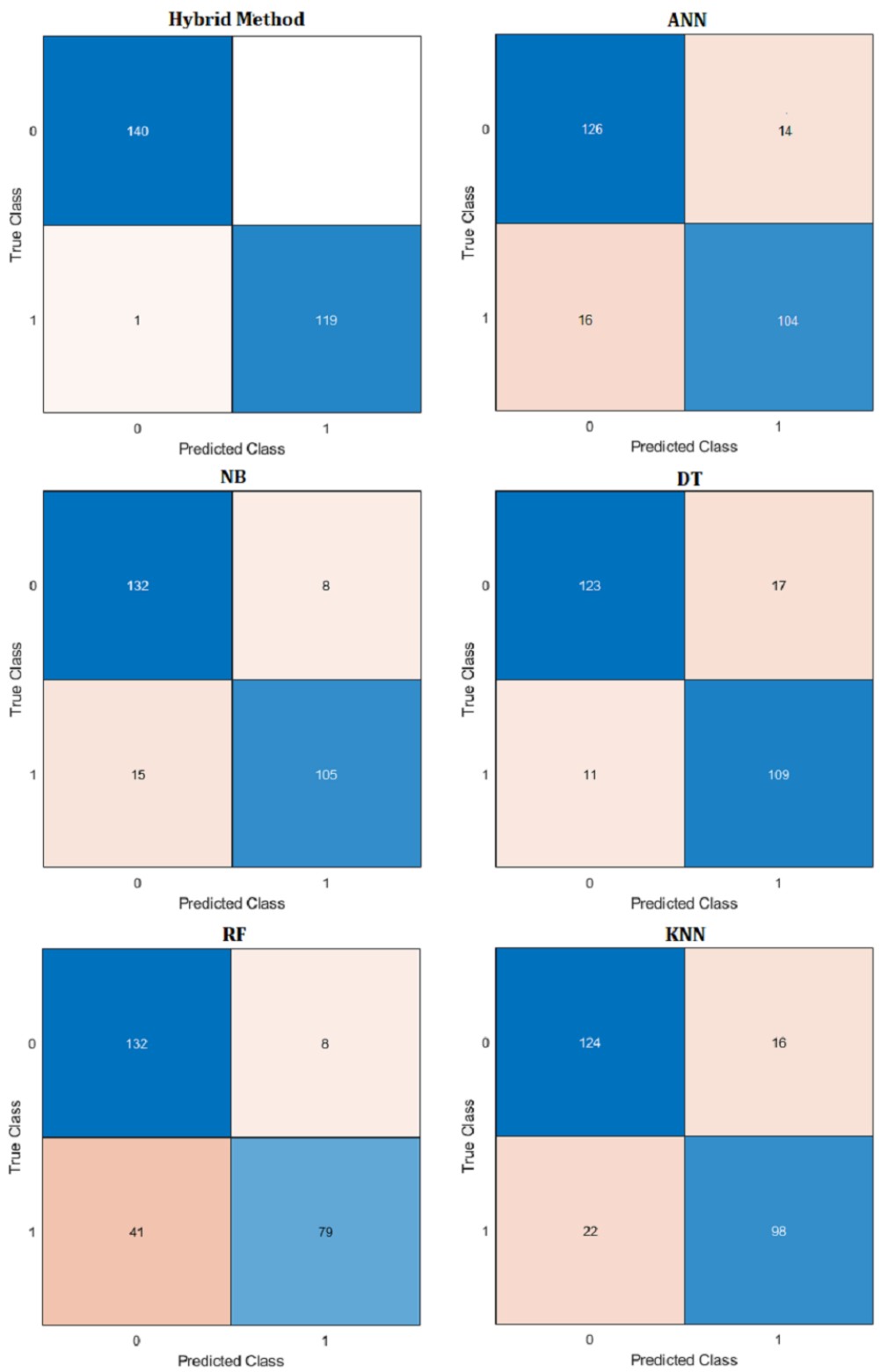

**Figure 14** Confusion matrix of hybrid method and ML algorithms.

**Table 9  Performance metrics of the hybrid method (%).**

|  | K-means | SVM | K-means vs SVM |
|---|---|---|---|
| Accuracy | 0.9063 | 0.9468 | 0.9936 |
| Recall | 0.9075 | 0.9295 | 0.9881 |
| F1 Score | 0.9165 | 0.9361 | 0.9912 |
| Precision | 0.9123 | 0.9286 | 0.9907 |

**Table 10  Hybrid method and ML performance metrics (%).**

|  | Hybrid | ANN | NB | DT | KNN | RF |
|---|---|---|---|---|---|---|
| Accuracy | 0.9963 | 0.8744 | 0.9129 | 0.8925 | 0.8543 | 0.8123 |
| Recall | 0.9775 | 0.8661 | 0.8890 | 0.8781 | 0.8361 | 0.7846 |
| F1 Score | 0.9823 | 0.8709 | 0.9007 | 0.8846 | 0.5438 | 0.7967 |
| Precision | 0.9765 | 0.8662 | 0.8890 | 0.8782 | 0.8361 | 0.7846 |

absolute percentage error (MAPE) prediction metrics and regression graphs are presented in Fig. 15.

# DISCUSSION AND LIMITATIONS

The main purpose of the study is to propose an ML-based hybrid decision support method for decision makers, free from administrative and software problems, based on quantitative criteria for determining the most appropriate risk assessment techniques for sectors among many factors specified in literature studies. In this way, decision makers will be able to choose the most appropriate technique according to restrictions, limitations and priorities among the available alternatives according to sectors or working conditions.

There are two main motivations for the study. The first of these is to determine the factors that are effective in risk assessment in order to recommend an appropriate risk assessment method. To do this, a comprehensive list must first be created by classifying the factors found in the pertinent literature. For this purpose, a comprehensive literature study and analysis of quantitative data were conducted. Thus, a quantitative data set to which ML algorithms can be applied was created. Unreliable results can arise from using qualitative methods to gather expert opinions based on subjective assessments (*Gupta & Clarke, 1996*). By balancing the level of precision, quantitative data measurement can improve the validity of the findings. One way to do this is to use ML algorithms, which is an artificial intelligence technique.

There are still certain research gaps even though the literature evaluation has identified a number of parameters influencing the choice of suitable risk assessment approaches. Risk assessment techniques should be compared and evaluated in accordance with these features, even though factors like system design (*Marhavilas, Koulouriotis & Gemeni, 2011*), diversity of risks and system complexity (*Moraru, Babut & Cioca, 2014*), and the presence of different parties in the project (*Dey & Ogunlana, 2004*) are crucial. It is challenging to grade. These variables can be applied to other phases of the risk assessment technique

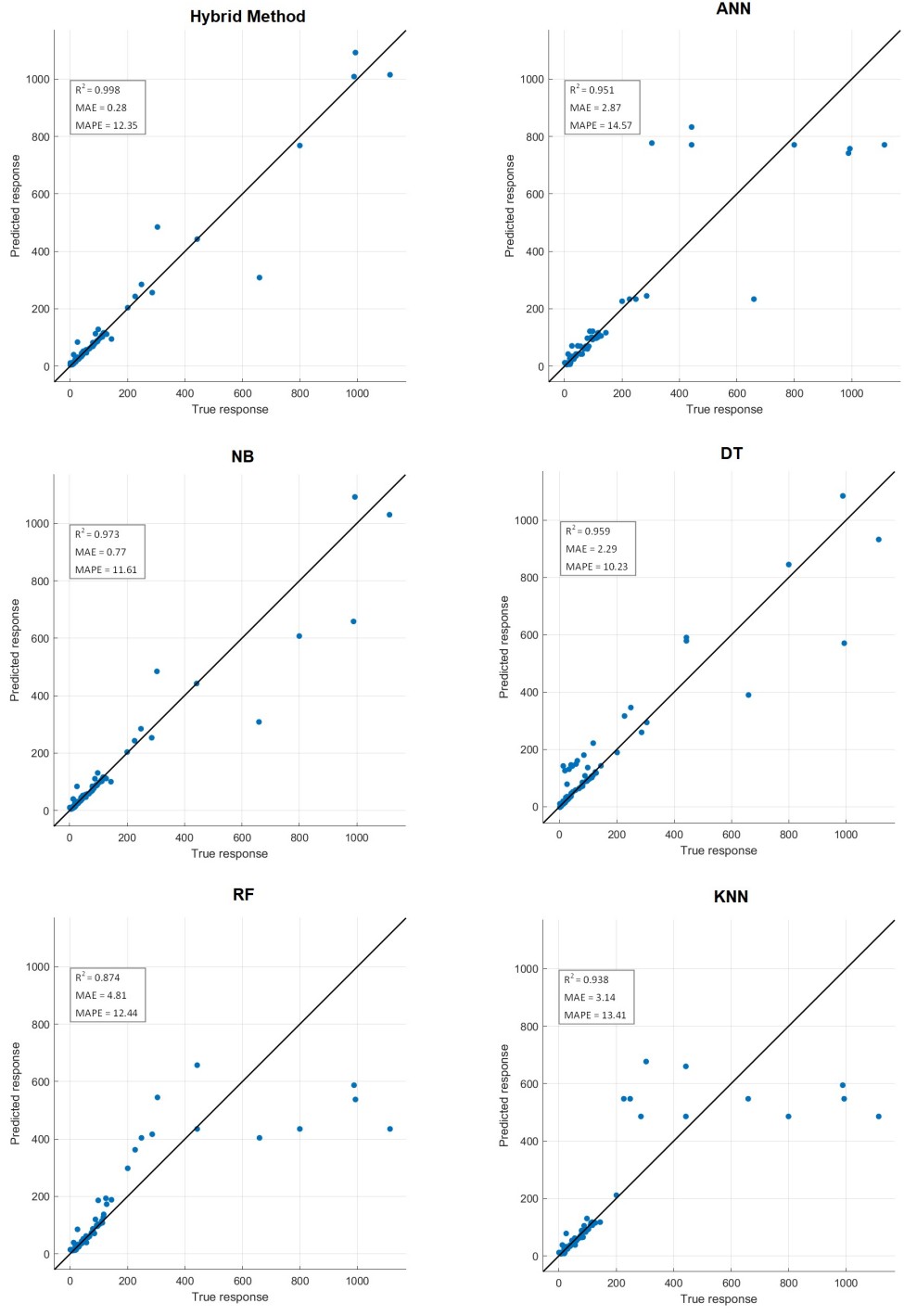

**Figure 15** **Prediction metrics of hybrid method and ML algorithms.**

selection process, like assessing the industry being studied or developing the backdrop for the risk assessment.

Certain studies lack defined criteria, and the usage of certain components is ambiguous due to imprecise definitions. Additionally, the selection criteria were determined using unreliable, simplistic procedures. A subjective choice will result from the application of some techniques, such as the Likert scale (*Karam, Hussein & Reinau, 2021*) or the solicitation of expert opinions without first establishing the qualifications.

Although the existing literature and theoretical discussions support the accuracy of the determined criteria, more comprehensive and effective results can be obtained with the participation of sector representatives, sector employees, academics, state and non-governmental representatives in the process. In addition, in the study, 10 features were evaluated with the chi square method for the most effective qualities, as well as expert opinions. The chi square method, also known as the X2 test, can be used to determine whether the variables are suitable for describing the data set (*Kavzoğlu, Şahin & Çölkesen, 2014*). The use of unimportant attributes that will not affect the result in the selection of the most appropriate risk assessment method increases the processing time and may also cause a decrease in performance (*Chebrolu, Abraham & Thomas, 2005*).

The second motivation of the study is the selection of risk assessment methods. The nature of the study, knowledge of the system, availability of quantitative data, and well-established sources were taken into account in the selection of these methods. When we look at the risk assessment methodologies, that is, methodologies and standards, all over the world within the scope of ISO 31010 Risk Management Standard for the selection of risk assessment methods, it can be seen that there are more than 150 methods. Within the scope of the study, the 10 most commonly used methods in both literature review and field applications were discussed.

There are various factors and criteria that differ from each other in choosing the most appropriate risk assessment method for a sector. Furthermore, selection criteria will differ from region to region in tandem with variations in the nature of risk (*Chemweno et al., 2015*). As a result, risk assessment strategies must be assessed and prioritized. The decision maker must be able to quickly identify the best technique and rank the techniques objectively, not simply subjectively (*Ford et al., 2008*; *Dey & Ogunlana, 2004*).

## CONCLUSION

This study provides practical information that enables the selection of the most appropriate risk assessment methods to reduce accidents occurring during work activities of both sectors and corporate organizations, reduce social and economic losses, and increase worker safety. In the study, a hybrid model was designed that enables the determination of the most appropriate risk assessment methods for sectors. In the model, k-means, which can show high performance in large data sets, is easy to implement and has little time complexity, and the SVM algorithm, which works well with a clear separation margin and high dimensional space, has no overfitting problems and can model complex boundaries with high accuracy, were used. A total of 26 sectors, 10 different risk assessment models,

and 10 criteria were determined for the study. In selecting the most appropriate risk assessment methods, the results of k-means, SVM, and hybrid method were examined and their performances were compared. The specific conclusions of this article are as follows:

- The literature on the selection of risk assessment methods in occupational health and safety practices has been enriched.
- The study, which covers all sectors for the first time in the literature and proposes the most appropriate risk assessment method for each sector, will contribute to reducing the OHS risk level, protecting the occupational health and safety of workers, and reducing material and moral losses in enterprises.
- The proposed model has an accuracy rate of 99.6%. In addition, the data set was trained with five different ML algorithms to make a more reliable and effective comparison and evaluation. Obtained results were 87.4% for ANN, 91.2% for NB, 89.2% for DT, 85.4% for KNN and 81.2% for RF.

In summary, the proposed ML-based hybrid method showed a higher performance than traditional clustering and classification algorithms. Instead of a single threshold value limitation, separate threshold values were determined for each subset according to their characteristics, eliminating the obligation and creating a flexible structure with values specific to the clusters. In the proposed model, the time complexity and memory requirement of the SVM algorithm were reduced by dividing the data set into smaller subsets with the k-means algorithm, which gives the fastest result. In the future, for the method to obtain more accurate results and have a wider scope of application, the number of features used in ML algorithms should be increased and the sectors should be made specific by dividing them into sub-sector units.

### Funding
The author received no funding for this work.

### Competing Interests
The author declares there are no competing interests.

### Author Contributions
- Fatih Topaloglu conceived and designed the experiments, performed the experiments, analyzed the data, performed the computation work, prepared figures and/or tables, authored or reviewed drafts of the article, and approved the final draft.

### Data Availability
The raw data and code are available in the Supplemental Files.

## Supplemental Information

Supplemental information for this article can be found online at http://dx.doi.org/10.7717/peerj-cs.2198#supplemental-information.

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
