# Peer review of "A hybrid approach based on k-means and SVM algorithms in selection of appropriate risk assessment methods for sectors"

_PeerJ Computer Science, doi:10.7717/peerj-cs.2198_

## Round 0.1 · original submission · Major Revisions

The study is interesting, but needs some modifications to increase the validity of the results.

I would recommend a more in-depth and detailed description of the experimental setup and results.

Furthermore, it would be interesting to more carefully describe the contribution and novelty of the study compared to the existing literature.

·

Basic reporting

The paper discusses the importance of selecting the appropriate risk assessment method (RAM) for a given work environment. With numerous RAMs available, choosing the right one is crucial. The study proposes a hybrid approach using k-means clustering and support vector machine (SVM) classification algorithms to determine the most suitable RAMs for different sectors. The approach involves dividing the dataset into subsets using k-means, running SVM on each subset, and combining the results. This creates a flexible structure with separate threshold values for each sub-cluster, providing machine support for selecting the most suitable RAMs. The proposed hybrid method outperforms other machine learning algorithms. I

Experimental design

The paper discusses the importance of selecting the appropriate risk assessment method (RAM) for a given work environment. With numerous RAMs available, choosing the right one is crucial. The study proposes a hybrid approach using k-means clustering and support vector machine (SVM) classification algorithms to determine the most suitable RAMs for different sectors. The approach involves dividing the dataset into subsets using k-means, running SVM on each subset, and combining the results. This creates a flexible structure with separate threshold values for each sub-cluster, providing machine support for selecting the most suitable RAMs. The proposed hybrid method outperforms other machine learning algorithms. I It Addresses a significant problem in risk assessment It Proposes a novel hybrid approach combining k-means and SVM and Demonstrates improved results compared to other machine learning algorithms
Weaknesses:
Limited explanation of the k-means and SVM algorithms
Unclear how the 26 sectors, 10 RAMs, and 10 criteria were selected
Lack of detailed results and performance metrics
Suggestions for Improvement:
Provide a more comprehensive explanation of the machine learning algorithms used
Clarify the selection process for sectors, RAMs, and criteria
Include more detailed results, performance metrics, and comparisons with other methods

Validity of the findings

The paper discusses the importance of selecting the appropriate risk assessment method (RAM) for a given work environment. With numerous RAMs available, choosing the right one is crucial. The study proposes a hybrid approach using k-means clustering and support vector machine (SVM) classification algorithms to determine the most suitable RAMs for different sectors. The approach involves dividing the dataset into subsets using k-means, running SVM on each subset, and combining the results. This creates a flexible structure with separate threshold values for each sub-cluster, providing machine support for selecting the most suitable RAMs. The proposed hybrid method outperforms other machine learning algorithms. I It Addresses a significant problem in risk assessment It Proposes a novel hybrid approach combining k-means and SVM and Demonstrates improved results compared to other machine learning algorithms There is limited explanation of the k-means and SVM algorithms
It is little unclear how the 26 sectors, 10 RAMs, and 10 criteria were selected.

Additional comments

The paper could have provided a more comprehensive explanation of the machine learning algorithms used and it would be great to Clarify the selection process for sectors, RAMs, and criteria
Include more detailed results, performance metrics, and comparisons with other methods

·

Basic reporting

1. Numerical results and values should be included in the abstract part.
2. Novel experimental work is missing
3. Results are not defined accurately.

Experimental design

poor experiment work

Validity of the findings

no novel work is find

Additional comments

Major revisions are required.

---

## Round 0.2 · accepted · Accept

The authors completed the revision of the manuscript based on suggestions provided by the reviewers. I confirm that all requests and comments from reviewers have been adequately addressed with valid additions and justifications.